

# Evaluation of FGFR1 as a diagnostic biomarker for ovarian cancer using TCGA and GEO datasets

Huiting Xiao[1,*], Kun Wang[2,*], Dan Li[1], Ke Wang[1] and Min Yu[1]

[1] Department of Gynecologic Oncology, Tianjin Medical University Cancer Institute and Hospital, National Clinical Research Center for Cancer, Tianjin Key Laboratory of Cancer Prevention and Therapy, Tianjin, China

[2] Department of Urologic Oncology, Tianjin Medical University Cancer Institute and Hospital, National Clinical Research Center for Cancer, Tianjin Key Laboratory of Cancer Prevention and Therapy, Tianjin, China

[*] These authors contributed equally to this work.

## ABSTRACT

**Background**. Malignant ovarian cancer is associated with the highest mortality of all gynecological tumors. Designing therapeutic targets that are specific to OC tissue is important for optimizing OC therapies. This study aims to identify different expression patterns of genes related to FGFR1 and the usefulness of FGFR1 as diagnostic biomarker for OC.

**Methods**. We collected data from The Cancer Genome Atlas (TCGA) and the Gene Expression Omnibus (GEO) databases. In the TCGA cohort we analyzed clinical information according to patient characteristics, including age, stage, grade, longest dimension of the tumor and the presence of a residual tumor. GEO data served as a validation set. We obtained data on differentially expressed genes (DEGs) from the two microarray datasets. We then used gene set enrichment analysis (GSEA) to analyze the DEG data in order to identify enriched pathways related to FGFR1.

**Results**. Differential expression analysis revealed that FGFR1 was significantly down-regulated in OC specimens. 303 patients were included in the TCGA cohort. The GEO dataset confirmed these findings using information on 75 Asian patients. The GSE105437 and GSE12470 database highlighted the significant diagnostic value of FGFR1 in identifying OC (AUC $= 1$, $p = 0.0009$ and AUC $= 0.8256$, $p = 0.0015$ respectively).

**Conclusions**. Our study examined existing TCGA and GEO datasets for novel factors associated with OC and identified FGFR1 as a potential diagnostic factor. Further investigation is warranted to characterize the role played by FGFR1 in OC.

## INTRODUCTION

Ovarian cancer is a common gynecological malignant cancer. Because its mortality rate ranks first among gynecological tumors, it has become a major health risk for women (*Chen et al., 2018*; *Meys et al., 2018*). The onset of ovarian cancer is usually hidden and most patients have no obvious symptoms in the early stages (*Partridge, Phillips & Menck,*

Corresponding authors
Ke Wang, kewang12@163.com
Min Yu, minyu816@163.com

*1996*). Therefore, at the time of diagnosis patients may have already reached relatively advanced stages of OC. The five-year survival rate of patients is approximately 40% (*Partridge, Phillips & Menck, 1996*; *Stewart et al., 2017*) while the overall five-year survival rate for patients whose cancer has reach an advanced stage is only 20–30% (*Bhatt et al., 2016*). The ovarian cancer patients who received guideline-based treatment have better five-year survival rate. However, even after initial treatment, most patients with epithelial ovarian cancer will experience a recurrence, and currently there is no cure (*Beltrame et al., 2015*; *Bruchim et al., 2016*; *Ganapathi et al., 2016*). Moreover, traditional treatment methods, including surgery and chemotherapy, do not significantly affect the survival rate of the disease. In consequence, the development of new methods to detect OC in the early stages of its progression as well as individualized treatment plans for ovarian cancer are key steps for improving clinical efficacy and safety.

The targeted treatment of OC plays an increasingly important role in the comprehensive treatment of OC and new tumor treatment strategies depend on the search for new targets (*Huang et al., 2018*; *Shivange et al., 2018*; *Villar-Prados et al., 2018*; *Zhai et al., 2018*). With the development of sequencing technology and acquirement of a large amount of biological data, bioinformatics can be used to understand and find new biomarkers of tumor (*Yang et al., 2020*; *Wu et al., 2019*; *Tao et al., 2019*). The Vascular Endothelial Growth Factor (VEGF) signaling pathway is essential for tumor angiogenesis (*Albini et al., 2018*; *Caron-Beaudoin, Viau & Sanderson, 2018*; *Lee et al., 2018a*; *Lee et al., 2018b*; *Lin et al., 2018*). Anti VEGF therapies clearly display anti-angiogenic efficacy in the treatment of pathological angiogenesis as well as cancers including lung cancer (*Kabbinavar et al., 2014*), glioma (*Griveau et al., 2018*), metastatic renal cell carcinoma (*Escudier et al., 2007*; *Grünwald et al., 2011*), metastatic colorectal cancer (*Bennouna et al., 2018*), and ovarian cancer (*González Martín et al., 2018*; *Lee et al., 2018a*; *Lee et al., 2018b*). Clinical data has shown that current anti-angiogenic targeted drugs have been successfully introduced in the anti-cancer therapy, but they still have more drawbacks (*Abdullah & Roman, 2012*; *Poletto et al., 2018*; *Wang et al., 2018*). VEGF is a crucial angiogenic factor. A lot of angiogenic factors, including angiopoietins, platelet derived growth factor (PDGF) and fibroblast growth factor (FGF), collaborate with VEGF in the angiogenic process (*Abdullah & Roman, 2012*; *Ip et al., 2018*; *Lieu et al., 2011*; *Semrad & Mack, 2011*).

The human fibroblast growth factor receptor (FGFR) family consists of four members: FGFR1 to FGFR4 and the native ligand of FGFRs is fibroblast growth factors (FGFs) (*Lemmon & Schlessinger, 2010*; *Weiner & Zagzag, 2000*). Dysregulation of FGFRs has been implicated in a wide variety of cancers, such as urothelial carcinoma, hepatocellular carcinoma, ovarian cancer and lung adenocarcinoma (*Cole et al., 2010*; *Fearon, Gould & Grose, 2013*; *Shuyan et al., 2019*). There is evidence that differential expression in specific FGFR may be related to prognosis or sensitivity to cancer treatments (*Turner et al., 2010*). At this time, several FGFR inhibitors are FDA approved for treatment of cancer, including lenvatinib approved for iodine-refractory, well-differentiated thyroid carcinoma, regorafenib approved for advanced colorectal carcinoma and drug-resistant gastrointestinal stromal tumors (GIST), ponatinib approved for drug-resistant chronic myelogenous leukemia (CML) and Philadelphia chromosome-positive acute lymphocytic

leukemia (ALL), and pazopanib approved for renal cell carcinoma and sarcoma. The FGF/FGFR-system plays a critical role in carcinogenesis, but little is known of its influence in ovarian cancer.

This study aims to provide insight into the differential expression of FGFR-linked genes in OC and normal tissues by comparing TCGA and GEO data. The detection of DEGs between OC and non-OC tissues may facilitate the identification of novel therapeutic targets of OC. To better understand the role of FGFR1 in ovarian cancer in individual studies, a meta-analysis was designed to synthesize current findings. Here, DEGs found by comparing gene expression in OC and non-OC samples were screened using the R software. The enrichment pathway analysis of DEGs was performed using GSEA. Our results suggest that the significant diagnostic ability of FGFR1, but also note that more evidence is needed to improve disease prognosis and to design effective drugs to treat ovarian cancer.

## METHODS

### TCGA data description

To obtain a dataset based on studies listed in The Cancer Gene Atlas (TCGA) database, patients' gene expression and clinical information were downloaded from the publicly available TCGA Data Portal at https://tcga-data.nci.nih.gov/tcga/. For the TCGA cohort, gene expression profiles were studied in 303 female patients with histologically confirmed ovarian cancer who had undergone prior surgical resection and received no pretreatment. Gene expression profiles were measured experimentally using an Illumina HiSeq2000 RNA Sequencing instrument at the University of North Carolina TCGA genome characterization center. Level 3 data was downloaded from the TCGA data coordination center. This dataset shows gene-level transcription estimates, as in $\log2(x+1)$ transformed RSEM normalized counts. Genes were mapped onto the human genome coordinates using HUGO probeMap. Reference to method description from University of North Carolina TCGA genome characterization center. All analyses were performed using FDR (False Discovery Rate) corrected $q$-values adjusted to 0.05.

### GEO dataset selection

We obtained OC microarray profiles from the GEO database (http://www.ncbi.nlm.nih.gov/geo/). The following keywords were used to query the GEO database: (ovary OR ovarium OR oophoron OR ootheca OR germarium) AND (cancer OR carcinoma OR tumor OR tumour OR neoplas* OR malignan*). All the functional genomics data of FGFR1 were requested and assembled from the GEO Database (http://www.ncbi.nlm.nih.gov/geo/) with the closing date of 10 February 2019. The inclusion and exclusion criteria of this study were as follows: (1) only human tissue studies were included, and samples from cell lines and animal models were excluded; (2) the expression data of the experimental and control groups must be provided or calculated; (3) dual-channel microarray studies were excluded; (4) DNA methylation arrays studies were excluded; (5) studies without a control group, as well as studies on other cancers were excluded; (6) studies with less than 20 cases were excluded. Differentially expressed genes (DEGs) identified by comparing expression in

ovarian cancer and normal samples were analyzed using the R language software. Samples with both corrected $P$-values <0.05 and log fold changes (FC) >1 were deemed to be DEGs.

## Gene set enrichment analysis (GSEA)

GSEA was performed using GSEA software 3.0 from the Broad Institute as previously described (*Subramanian et al., 2005*). GSEA evaluates gene expression profiles from samples belonging to normal and tumorous samples, and analyzes data in terms of gene sets. Normalized enrichment scores (NES) were acquired by analyzing genes with permutations 1,000 times. A gene set was considered to be significantly enriched if it had a normal $p$-value <0.05. However, if the number of samples per group was fewer than 7, gene set was selected as permutation type, and FDR <0.05 was set as the criterion of statistical significance.

## Statistical analysis

Overall survival was calculated from the data of surgery to the date of death or last follow-up. Recurrence-free survival was defined as the time from the date of surgery to the date of first recurrence or last follow-up. Patients without events or death were recorded as censored at the time of last follow-up. Pathological stage and grade were considered to be distinct categorical variables. Age, longest dimension and residual tumor were included in the model as continuous variables. Gene expression levels from the TCGA database were included as continuous variables.

All data are displayed as mean $\pm$ standard deviation (SD) for each group. Student's $t$-tests were performed to test for differences in mean values of vaviables of interest between two groups, whereas one-way analyses of variance (ANOVAs) were used to test for differences in means among three or more groups. Standardized mean difference (SMD) was applied to evaluate the association between FGFR1 levels and OC using RevMan 5.3.0. We pooled SMDs across GEO datasets using the Mantel-Haenszel formula (fixed-effect model) or the DerSimonian-Laird formula (random-effect model). A fixed-effect model was adopted when the Q statistic was considered significant ($p > 0.1$, or $I^2 < 50\%$); otherwise, a random-effect model was used.

# RESULTS

## Study Characteristics

The present study consists of several processes sequentially (Fig. 1), that is, TCGA-based RNA-seq data aggregation and clinical values, GEO-based data verification, meta-analyses based on GEO and TCGA, and multiple bioinformatics analyses. A total of twelve GEO datasets (GSE105437, GSE66957, GSE66387, GSE40595, GSE29450, GSE29156, GSE27651, GSE26712, GSE18521, GSE18520, GSE17308, and GSE12470) were collected for use in our study. Two datasets (GSE105437 and GSE12470) contained detailed information on Asian populations, and these were used in this study. The platform for GSE105437 was GPL570, the [HG-U133_Plus_2] Affymetrix Human Genome U133 Plus 2.0 Array, includes 5 normal ovarian surface epithelial samples and 10 high-grade stage III invasive serous ovarian cancer samples. The platform for GSE12470 was GPL887, the Agilent-012097

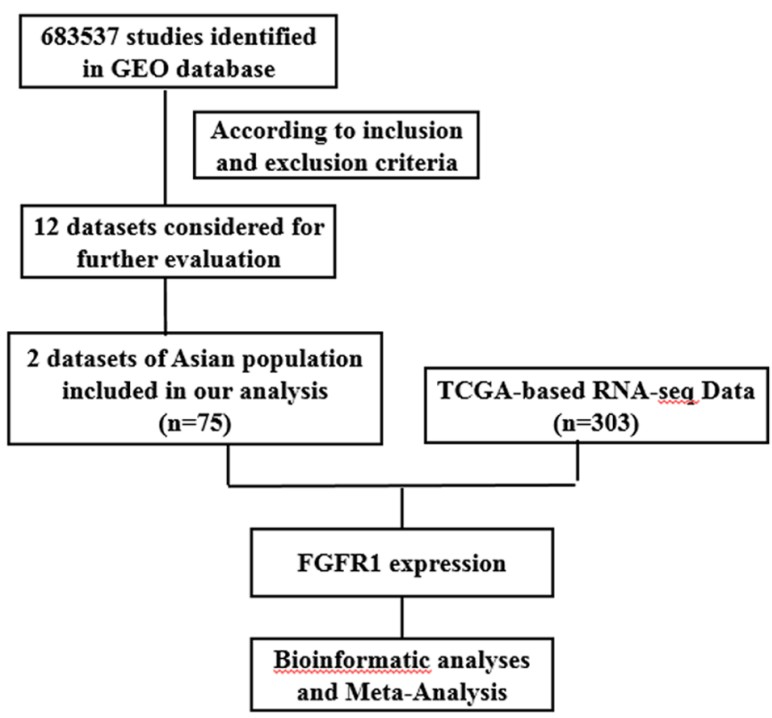

**Figure 1** Flow chart of study selection for GEO and TCGA based data.

Human 1A Microarray (V2) G4110B; this platform, included 10 normal peritoneum samples, 35 advanced stage serous ovarian cancer samples and 8 early stage serous ovarian cancer samples. In addition, 308 cases with EGFR or FGFR expression data were screened in TCGA database excluding 5 cases with recurrence, and 303 patients form this database were included in this study. A flowchart and other detailed information regarding the studies included in the meta-analysis are shown in Fig. 1.

## FGFR expression in OC based on GEO

FGFR or EGFR expression was initially assessed in a series of ovarian tumor and normal tissues based on GEO dataset (Fig. 2). The datasets were analyzed with GeneSpringGX and mapped with Graphpad 6.0 software. The expression levels of FGFR1 in ovarian cancer tissues were significantly lower than in non-cancer control tissues in both GSE105437 and GSE12470 with analyses performing Fold change cut-off adjusted to 2.0 and corrected $p$-values cut-off adjusted to 0.05($p < 0.0001$ and $p = 0.0032$, respectively; Figs. 2A and 2B). The expression of FGFR1 was significantly different in advanced ovarian cancer, but not in early ovarian cancer in GSE12470 ($p = 0.0124$; Fig. 2C). The expression levels of EGFR in ovarian cancer tissues were significantly lower than in non-cancer control tissues in GSE105437 ($p = 0.0008$; Fig. 2D), whereas no difference was found in GSE12470. The expression levels of FGFR2 and FGFR3 were significantly different only in GSE12470 ($p = 0.0046$ and $p = 0.0049$, respectively; Figs. 2E and 2F). Therefore, FGFR1 was chosen for subsequent analysis.

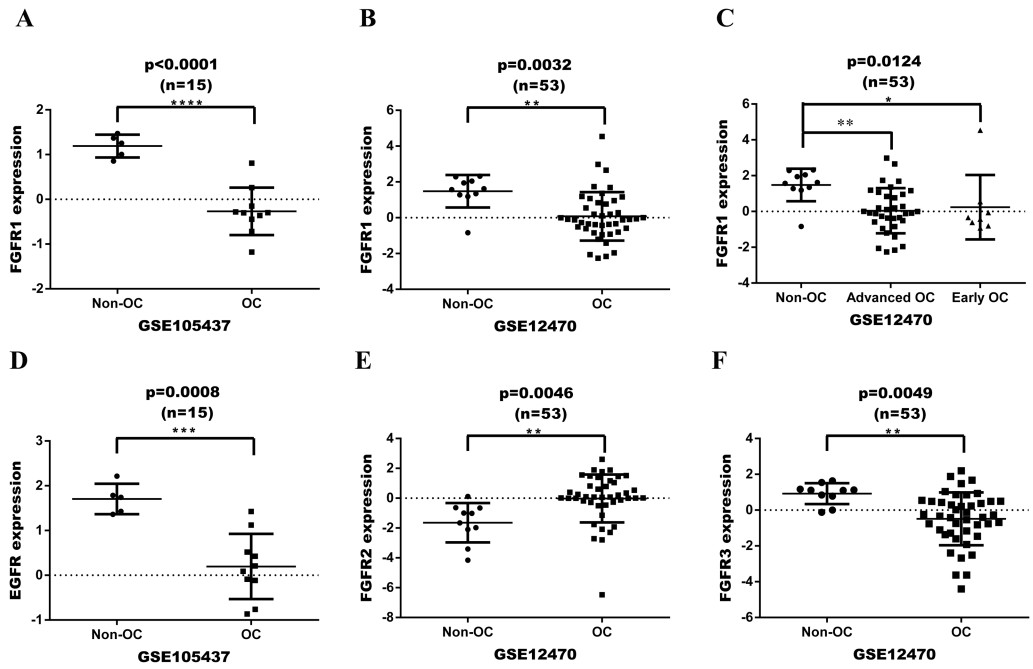

**Figure 2 Expression date in OC tissue from the GEO dataset.** (A) differential expression of FGFR1 in GSE105437; (B) differential expression of FGFR1 in GSE12470; (C) differential expression of FGFR1 in GSE12470 with subgroup analysis; (D) differential expression of EGFR in GSE105437; (E) differential expression of FGFR2 in GSE12470; E differential expression of FGFR3 in GSE12470.

To have a more comprehensive understanding of the biomedical predictive value of FGFR1, ROC Curves were provided to investigate the diagnostic value of GSE105437 and GSE12470 in distinguishing OC tissues from normal controls. As shown in Fig. 3, the Area Under the Curve (AUC) of GSE105437 outperformed GSE12470 (AUC=1, $p = 0.0009$ and $AUC = 0.8256$, $p = 0.0015$ respectively). Therefore, we considered FGFR1 might play an important role in diagnosing ovarian cancer. In brief, no significant differences were found between OC and nontumor groups based on the GSE105437 and GSE12470 data considered here (SMD $= -6.22$; 95% CI, $-7.48$ to $-4.96$; $p = 0.93$, Fig. 4A). Moreover, we found no significant heterogeneity by fixed-effected model ($p < 0.00001$, $I^2 = 0\%$). Finally, the funnel plot shown in Fig. 4B did not imply significant publication bias.

## Identification of DEGs in ovarian cancer using integrated bioinformatics

The two ovarian cancer gene expression microarray datasets GSE105437 and GSE12470 were analyzed using the R *limma* package and were sorted according to logfold-change values with corrected *p*-values <0.05. The results of DEGs from the two databases showed in Fig. 5, and FGFR was identified both in the down-regulated genes (Green points).

## GSEA enrichment analysis of DEGs

To find enriched pathways related to FGFR1 and to identify its potential function, we performed a GSEA (Fig. 6). First, common pathways were found by comparing the

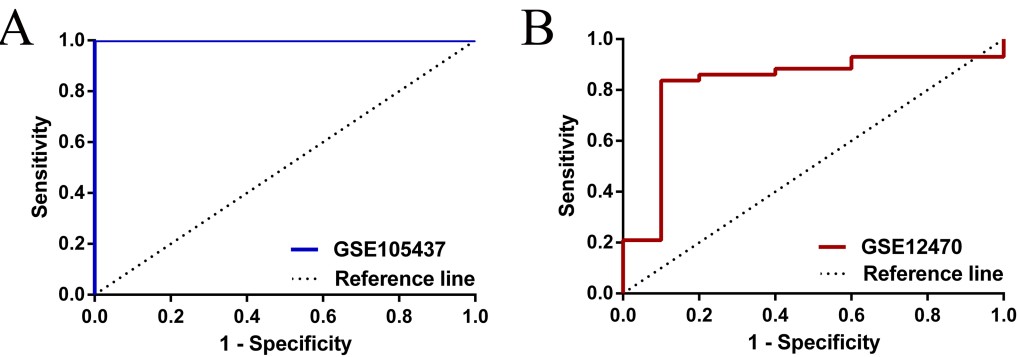

**Figure 3  ROC analysis of GSE105437 and GSE12470 of FGFR1 for the diagnosis of OC.** (A) GSE105437: The area under the ROC curve (AUC) 1.000, $p = 0.0009$; (B) GSE12470: AUC 0.8256, $p = 0.0015$.

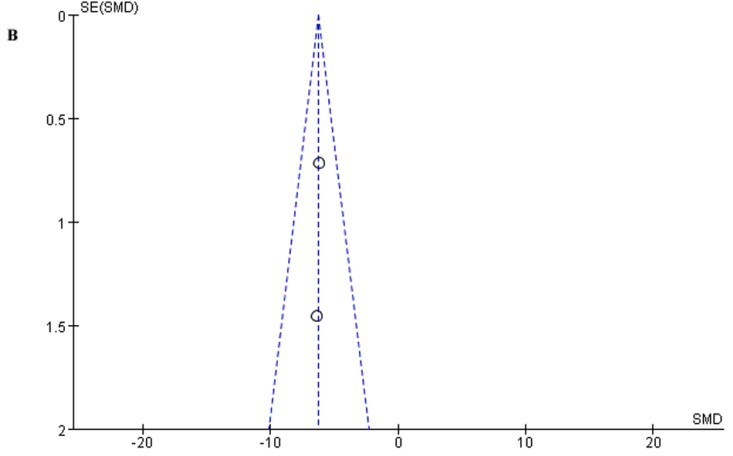

**Figure 4  Meta-analysis of the combined SMD for FGFR1 expression between OC and normal groups in the GEO database using the fixed effects models.** (A) Forest plot; (B) funnel plot.

GSE105437 (Fig. 6A) and GSE12470 (Fig. 6B) datasets. Although, many of these common pathways were not significantly associated with FGFR1, we found that the KEGG adherent junction signaling pathway was significantly correlated with FGFR1 in the GSE105437 dataset (Fig. 6C).

## Clinical characteristics of patients from the TCGA

The TCGA cohort consisted of 303 female patients with histologically confirmed ovarian cancer who had undergone prior surgical resection and received no pretreatment. Summary

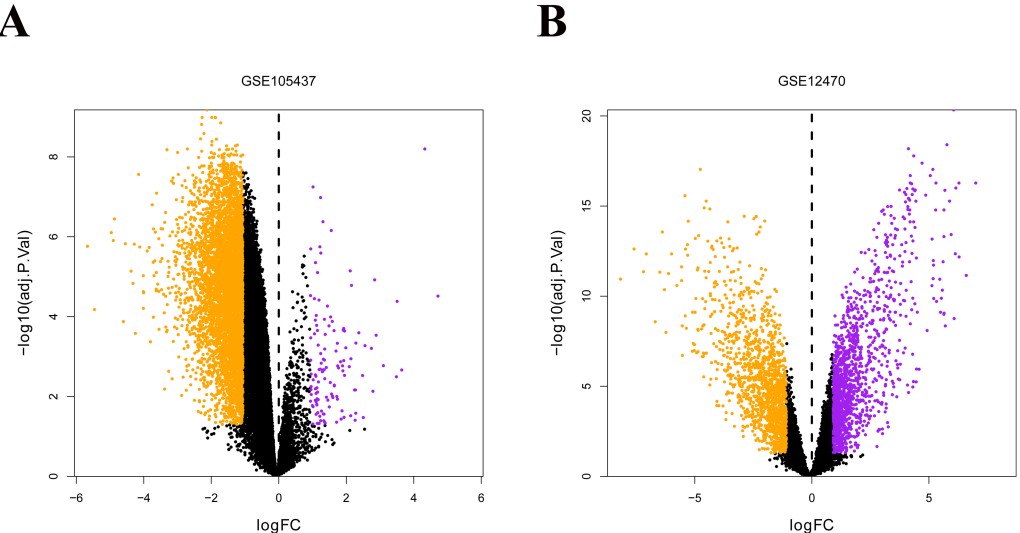

**Figure 5** **Differential expression of data between two sets of samples.** (A) GSE105437 (B) GSE12470. Red points represent upregulated genes (i.e., |FC| > 2.0 and a corrected *P*-value of < 0.05). Green points represent downregulated genes(i.e., |FC| > 2.0 and a corrected *P*-value of < 0.05). Black points represent genes with no significant difference in expression. FC is the fold change.

data of clinical indicators such as age, stage, grade, the longest dimension of the tumor, ANS, and residual tumor were shown in Table 1. We tested whether these variables were correlated with the relative expression of FGFR1 mRNA in OC tissues. We found no significant differences in the clinical features for all the tested parameters. The median follow-up time was 26.11 months and 182 patients died during follow-up. The results of the overall survival (OS) analyses, as calculated using the Kaplan–Meier method are shown in Table 2 and Fig. 7; we found significant differences in survival between all groups. We combined stage I and II into a single group, as well as grades G1 and G2 into another group for statistical analysis, since only 1 patient was included in the dataset for both stage I and grade G1. Residual tumors were found to be independently associated with increased OS according to univariate analysis ($p = 0.001$). In contrast, we found no statistical evidence that other characteristics, including age, stage, grade, tumor longest dimension, and ANS were independent prognostic factors. In addition, our results showed that patients with larger residual tumors, high FGFR expression, and advanced cancer stages were associated with shorter OS ($p = 0.001$, $p = 0.233$, $p = 0.148$, respectively; Fig. 7).

## DISCUSSION

Ovarian cancer is the deadliest gynecological malignant tumor due to the fact that this cancer is associated with delayed diagnosis, recurrence, and metastasis (*Chen et al., 2018*; *Meys et al., 2018*; *Partridge, Phillips & Menck, 1996*). Targeted treatments and immunohistochemistry offer the hope of improved treatments for OC patients in the future. Moreover, identifying biomarkers for targeted treatments is important for effective cancer diagnosis and treatment (*Shivange et al., 2018*; *Villar-Prados et al., 2018*). RNA

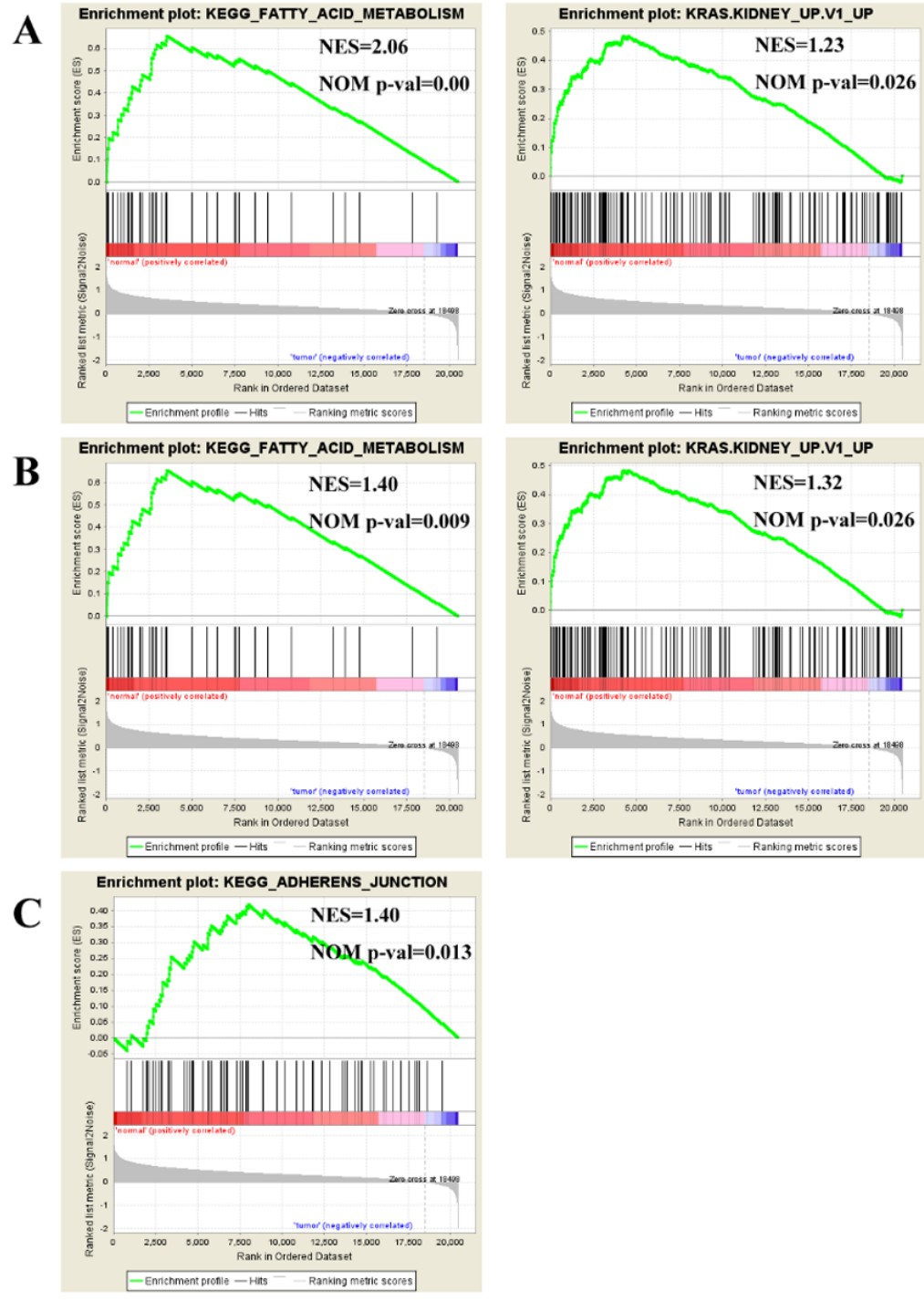

**Figure 6  Enrichment plots from gene set enrichment analysis (GSEA).** Common pathways in the GSE105437 (A) and GSE12470 (B) datasets according to GSEA, specific pathways involved FGFR1 GSE105437 datasets according to GSEA (C). ES, enrichment score; NES, normalized ES; NOM p-val, normalized *p*-value.

**Table 1  Correlations between the relative expression of FGFR1 mRNA in OC tissues and patient clinical indicators.**

| Clinicopathological parameters | N | Relative expression of FGFR1 mRNA | P value |
|---|---|---|---|
| Age | | | |
| ≤40 | 10 | 11.45 ± 0.3398 | |
| 40–60 | 157 | 11.60 ± 0.0887 | 0.4750 |
| ≥60 | 136 | 11.75 ± 0.1039 | |
| Stage | | | |
| I & II | 24 | 11.59 ± 0.2201 | 0.7667 |
| III & IV | 279 | 11.67 ± 0.06965 | |
| Grade | | | |
| G1+ G2 | 34 | 11.59 ± 0.1480 | |
| G3+ G4 | 261 | 11.67 ± 0.0741 | 0.8451 |
| Gx | 8 | 11.94 ± 0.3241 | |
| Longest dimension | | | |
| ≤1 cm | 92 | 11.60 ± 0.1201 | |
| 1–2 cm | 169 | 11.63 ± 0.0881 | 0.2510 |
| ≥2 cm | 42 | 11.93 ± 0.1843 | |
| ANS | | | |
| Bilateral | 209 | 11.63 ± 0.0783 | 0.5104 |
| Unilateral | 94 | 11.73 ± 0.1246 | |
| Residual tumor | | | |
| 0 | 95 | 11.72 ± 0.1156 | |
| ≤1 cm | 133 | 11.59 ± 0.0973 | 0.6568 |
| 1–2 cm | 23 | 11.88 ± 0.2868 | |
| 2 cm | 52 | 11.62 ± 0.1671 | |

sequencing is an accurate method used to identify such biomarkers (*Coenen-Stass et al., 2018*; *Jiang et al., 2018*). Thus, RNA-sequencing will likely become more important during patient treatment, both for predicting the efficacy of different therapies as well as in monitoring toxicity.

FGFR is a promising cancer biomarkers and has a canonical tyrosine kinase receptor structure (*Lieu et al., 2011*; *Semrad & Mack, 2011*). The FDFR family contains four members, FGFR1, FGFR2, FGFR3, and FGFR4 (*Eswarakumar, Lax & Schlessinger, 2005*). FGFR signaling is initiated by interaction with 23 different FGF ligands, and is involved in many cellular processes, including proliferation, differentiation, migration, survival, organogenesis, angiogenesis, and embryonic development (*Eswarakumar, Lax & Schlessinger, 2005*; *Klint & Claesson-Welsh, 1999*; *Ornitz & Itoh, 2015*; *Touat et al., 2015*). Recently, several mutations and alterations in FGFRs have been reported in cancers. In addition, specific alterations of FGFR have been found to be more frequent in certain types of tumors, thus making FGFR a suitable biomarker (*Brooks, Kilgour & Smith, 2012*; *Ornitz & Itoh, 2015*). However, the relationship between FGFR1 and ovarian cancer has not been well determined.

**Table 2  Univariate analyses of survival of 303 OC patients (using the Kaplan-Meier method).**

| Factor | n | 3-year survival(%) | $X^2$ | P |
|---|---|---|---|---|
| Age(years) | | | 6.463 | 0.04 |
| $\leq$40 | 10 | 71.4 | | |
| 40–60 | 157 | 70.5 | | |
| $\geq$60 | 136 | 51.2 | | |
| Stage | | | 2.088 | 0.148 |
| I & II | 24 | 77.9 | | |
| III & IV | 279 | 60.9 | | |
| Grade | | | 2.547 | 0.28 |
| G1+ G2 | 34 | 78.1 | | |
| G3+ G4 | 261 | 59.0 | | |
| Gx | 8 | 87.5 | | |
| Longest dimension | | | 0.47 | 0.79 |
| $\leq$1 cm | 92 | 59.7 | | |
| 1–2 cm | 169 | 65.4 | | |
| $\geq$2 cm | 42 | 51.8 | | |
| ANS | | | 0.574 | 0.449 |
| Bilateral | 226 | 62.7 | | |
| Unilateral | 77 | 58.3 | | |
| Residual tumor | | | 16.670 | 0.001 |
| 0 | 95 | 74.9 | | |
| $\leq$1 cm | 133 | 61.1 | | |
| 1–2 cm | 23 | 50.5 | | |
| 2 cm | 52 | 50.0 | | |
| FGFR1 | | | 1.425 | 0.233 |
| High expression | 152 | 58.2 | | |
| Low expression | 151 | 65 | | |

In this study, we examined FGFR1 expression in OC by examining RNA expression profiles in cancerous tissues sourced from the TCGA and GEO datasets. In samples from the TCGA database, we identified the clinical characteristics of OC patients that were found to be associated with FGFR1 expression. However we were not able to draw a strong functional link between FGFR1 and OC, since the TCGA dataset contained too fewer noncancerous samples, thus reducing the statistical power of the analysis. To gain insight into the functional molecular pathways of FGFR1 that are implicated in the development and progression of ovarian cancer, two samples from the GEO database (datasets GSE105437 and GSE12470, containing 53 OC samples and 15 non-OC samples) were subjected to further examination. No clinical data was available for these datasets, and therefore we were not able to corroborate our previous finding. Next, we investigated the expression patterns of FGFR1 in ovarian cancer samples from both datasets. Survival analysis showed that low expression of FGFR1 was closely associated with poor OS. Moreover, using data from the GEO datasets, we also found that FGFR1 was downregulated in ovarian cancer samples

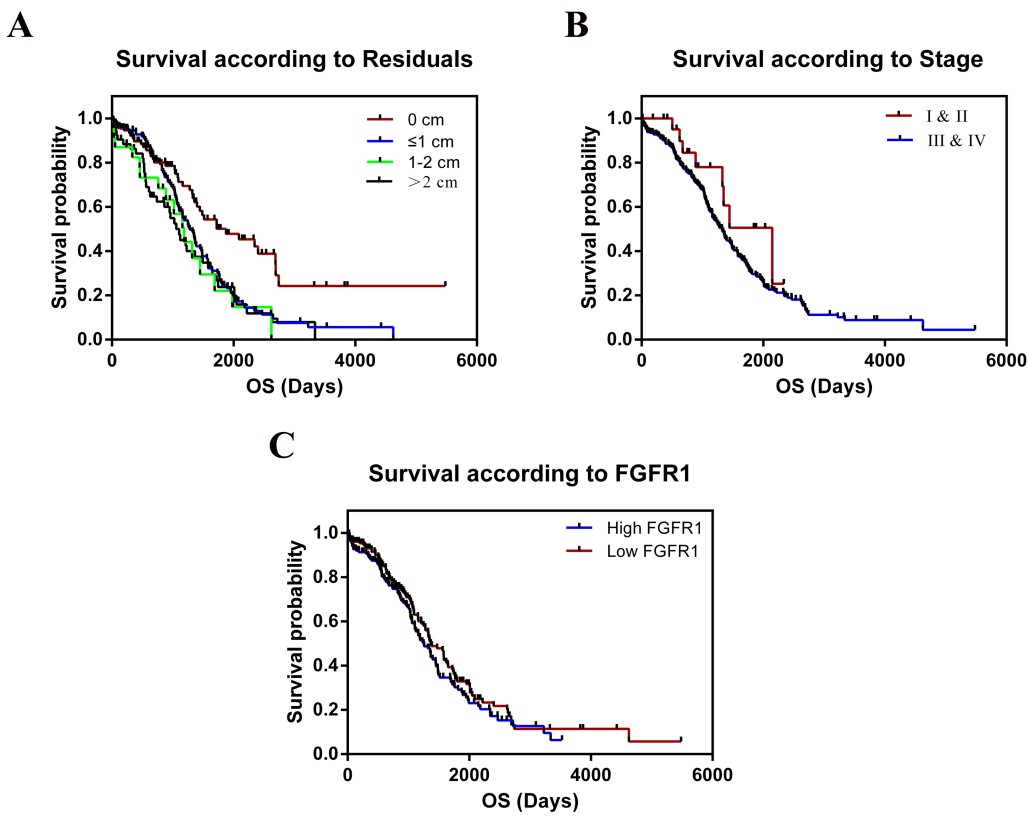

**Figure 7** **Kaplan–Meier estimates of OC overall survival of by residual tumor, stage and expression of FGFR1.** (A) Residual tumor, $p = 0.001$; (B) Stage, $p = 0.148$; (C) Expression of FGFR1, $p = 0.233$.

relative to normal tissues. However, no evidence indicated that FGFR1 is an independent factor affecting the clinical outcomes of ovarian cancer patients. Taken together, the results of our investigation, we found that FGFR1 might have significant diagnostic value in predicting OC for Asian populations (i.e., for the GSE105437 dataset: AUC = 1, $p = 0.0009$ and GSE12470 dataset: AUC = 0.8256, $p = 0.0015$). These results suggest that FGFR1 may be a very good diagnostic biomarker but not a good prognosis marker. More samples should be examined to validate these findings.

## CONCLUSIONS

In conclusion, this is a preliminary study designed to investigate the role played by FGFR1 in ovarian cancer. The meta analyses of ovarian cancer that we report here significantly extends our knowledge base to cognize that FGFR1 may be a good diagnostic biomarker in Asian populations (Fig. 3). In the future, these findings should be validated using new datasets, and immunohistochemistry or western blotting should be used to confirm transcriptomic results by examining protein contents. Finally, research into the value

of FGFR1 in cancer diagnosis should be performed in order to assess its usefulness for potential clinical application.

### Funding

This work was supported by the National Natural Science Foundation of China (NO. 81602299), Tianjin Health and Family Planning Commission Foundation of Science and Technology (NO. 16KG129) and National Natural Science Foundation of China (NO. 81602245). The funders had no role in study design, data collection and analysis, decision to publish, or preparation of the manuscript.

### Grant Disclosures

The following grant information was disclosed by the authors:
National Natural Science Foundation of China: 81602299.
Tianjin Health and Family Planning Commission Foundation of Science and Technology: 16KG129.
National Natural Science Foundation of China: 81602245.

### Competing Interests

The authors declare there are no competing interests.

### Author Contributions

- Huiting Xiao and Kun Wang performed the experiments, analyzed the data, authored or reviewed drafts of the paper, and approved the final draft.
- Dan Li performed the experiments, prepared figures and/or tables, and approved the final draft.
- Ke Wang performed the experiments, authored or reviewed drafts of the paper, and approved the final draft.
- Min Yu conceived and designed the experiments, performed the experiments, prepared figures and/or tables, authored or reviewed drafts of the paper, and approved the final draft.

### Data Availability

Data is available at NCBI GEO: GSE105437, GSE66957, GSE66387, GSE40595, GSE29450, GSE29156, GSE27651, GSE26712, GSE18521, GSE18520, GSE17308, and GSE12470.

### Supplemental Information

Supplemental information for this article can be found online at http://dx.doi.org/10.7717/peerj.10817#supplemental-information.

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
