# Peer review of "Evaluation of FGFR1 as a diagnostic biomarker for ovarian cancer using TCGA and GEO datasets"

_PeerJ, doi:10.7717/peerj.10817_

## Round 0.1 · original submission · Major Revisions

Dear Dr. Yu,
Thank you for your submission to PeerJ.

It is my opinion as the Academic Editor for your article - Evaluation of FGFR1 as a diagnostic biomarker for ovarian cancer using TCGA and GEO datasets - that it requires a number of Major Revisions.
My suggested changes and reviewer comments are shown below and on your article 'Overview' screen.

Along with the other issues, the key points you must focus are:
-Language and figures quality needs improvement
-More clarity of the research design/methodology is required.
-TCGA and other data needs clear description
-Biological insights out of this analysis is required

Please address these changes along with each reviewer's comments and resubmit. Although not a hard deadline please try to submit your revision within the next 55 days.

Thank you,
Best regards,
Debmalya Barh, PhD

Reviewer 1 ·

Basic reporting

- The use of language needs to be improved. There contains some grammatical errors, as well as some jargon.
- Quality of figures need to be improved.
- The authors should provide raw data, processed data and source codes for reproducing their results
- It is important to show some literature reviews related to the relevance of EGFR1 on ovarian cancer.

Experimental design

- Research question is a bit confusing. The authors have not shown the relationships between FGFR1 and ovarian cancer in the literature review. If it is not reported in previous works, why did the authors want to perform a work with this specific gene? At least it must have some experimental evidences.
- The authors have not explained well on their methods. It is necessary to make the methodology clearly, and the readers could reproduce their results.

Validity of the findings

If the authors could find that FGFR1 is an important biomarker for ovarian cancer and it is the first study shows this, it is very impact and novel. However, the current experiments and performance results did not show enough evidences on this point. For example:
- The survival analysis is not significantly different between high and low level of FGFR1, it cannot be convinced that EGFR1 is important in OC.
- How can we know the different distribution between training and validation cohort? The authors should perform some statistical tests on this.
- Why did the authors plot ROC curve for GEO datasets, but not in TCGA?

Additional comments

- Section 2.2, line 101 to 105, what is "(González Martín A OA)"?
- ROC curve or AUC has been used in previous works such as https://doi.org/10.1016/j.chemolab.2019.103855, https://doi.org/10.1016/j.neucom.2019.09.070, and https://doi.org/10.1186/s12859-019-2972-5. Therefore, the authors should refer more works in this description.
- Why were the normalization values similar between all samples in Fig. 3?

Reviewer 2 ·

Basic reporting

NA

Experimental design

NA

Validity of the findings

NA

Additional comments

In this manuscript “Evaluation of FGFR1 as a diagnostic biomarker for ovarian cancer using TCGA and GEO datasets “ Xiao et al. have discussed diagnostic value of FGFR expression in ovarian cancer patient cohort from TCGA and GEO datasets. Although authors have performed some bioinformatics analysis but haven’t discussed any biological insights gained from such analysis. This manuscript is very poorly written, no methodological details were provided and does not add any value to move the oncology field forward.

Please see the comments below:

1. Authors should clearly discuss the rationale of analyzing FGFR expression in ovarian cancer patients? Why FGFR1 and not FGFR2-4?

2. It is not clear how FGFR1 expression was predictive of survival. It is not shown in Figure1 of the manuscript. A quantitative way to subset patients in FGFR1 expression high or low patient groups should be used to analyze whether FGFR1 expression is predictive of survival in OV cancer patients. Does other FGFR family genes (e.g. FGFR2/FGFR3/FGFR4) are also predictive of survival?

3. Figure1 should indicate p-values and hazard ratio statistics. This should also be validated across various ovarian cancer cohort datasets (please see cbioportal – OV data). Compare FGFR1 expression levels in OV and normal cancer patients.

4. Methods section: TCGA data- authors should clearly describe TCGA data statistics and mention how many samples were downloaded from TCGA ovarian cancer data. It is not clear what analysis was performed using FDR cutoff (Line 94). Please clearly mention the details.

5. Figure5: Please explain what forest and funnel plots are representing here? What do authors meant by “we found no significant heterogeneity by fixed-effected model “ what is publication bias? (line 180-182). Section 3.4 describes methods and no results.

6. Section 3.2 : what does authors meant by : FGFR expression was validated in OC and normal ovary tissues in GEO dataset””. This section should be moved to methods as it doesn’t discuss any results

7. Figure2: This flowchart is not informative unless authors mention all the details. What was inclusion and exclusion criterion? What is the significance of mentioning all the GEO hits unless they were used for the data analysis.

8. Figure quality is very poor and mention statistics test used to calculate p-values.

9. Figure3: What does x and y axis represents? It is not clear how the data normalization was performed?

10. Figure 4: What is the significance of AUC curves here? It is not clear what authors wants to show here? Why this analysis was not carried out using TCGA data,

11. Figure legends are very poorly written and doesn’t describe the data shown in figures.

12. boxplots will help in understanding the variations in the clinical features as discussed in result section 3.1


13. Figure6: Mark where all the interesting genes lie.

14. Manuscript is very poorly written and has many spelling mistakes.

15. Line 115: samles ->”samples”

16. Figure 4: “Expression date” -> “Expression data”

Reviewer 3 ·

Basic reporting

This manuscript uses the datasets/databases for providing insights for using FGFR1 as a reliable marker for ovarian cancer. Overall the manuscript is very well written.

-Please provide more information on why FGFR linked genes were chosen for this study. This information should be included in the introduction section.

-The scientific language used to describe the work can be improved to make the statements more impactful. Some examples where the sentences can be modified include lines 44, 45, 66-70, 71-73, 77, 78, 93, 94, 139, 140.

Experimental design

- It has been mentioned in the discussion section that due to lack of control samples the statistical power of analysis was reduced. If possible, the authors should try to gain some more data for control set to strengthen the manuscript.

-The statements in “Conclusion” section overlap with the “Discussion” section. Authors should make the necessary changes.

- It is unclear what numbers (5) and (6) are used for in 106th line?

- Please remake Fig 1 and Fig 7. The figure quality is very poor and text is hardly visible.

Validity of the findings

Ovarian cancer is one of the hardest to treat cancers and the discovery of novel biomarkers will enable the development of drugs to treat cancer. Evaluation of FGFR1 as a prognostic marker in OC is important however more metadata from multiple databases are required to validate the findings of the current study. Nevertheless, the current study and data are important for the field.

---

## Round 0.2 · Minor Revisions

Dear Dr. Yu,

Thank you for your submission to PeerJ.

There are still few concerns on the revised version as raised by the Reviewer-2. I also feel the Language should be improved and the statements that are presumed and not proved in the work should be carefully handled.

Please revise the article accordingly and resubmit.

Thank you,
Best regards,
Debmalya Barh, PhD

Reviewer 1 ·

Basic reporting

No comment

Experimental design

No comment

Validity of the findings

No comment

Additional comments

Thanks for addressing my previous comments, but I think there are some comments that have not yet been addressed or not yet addressed well. I suggest a substantial revision on some unanswered points:
- The use of English still needs to be improved.
- More literature review should be added related to EGFR1 and ovarian cancer.
- I suggested the authors clearly revised the methodology part, but they mentioned that they revise the 'Study Characteristics' which located in "Results".
- I suggested the authors refer more works for ROC curve & AUC, but the authors might skip it. Please go back and re-check my previous comment.
- "Although FGFR1 expression is not significantly different in the survival analysis, we believed FGFR1 plays an important role in diagnosing ovarian cancer." --> this finding could not be convinced in their analyses.

Reviewer 3 ·

Basic reporting

no comment

Experimental design

no comment

Validity of the findings

no comment

Additional comments

Thanks for addressing the comments.

---

## Round 0.3 · accepted · Accept

Dear Dr. Yu,

Thank you for the revised manuscript. I am happy to accept the article now.

Wishing You and your Colleagues a very HAPPY NEW YEAR -2021

Thank you,
Best regards,
Debmalya Barh, PhD